# Two Sides of a Coin: Parental Disease-Specific Training as Seen by Health Care Practitioners and Parents in Pediatric Liver Transplantation

**DOI:** 10.3390/children8090827

**Published:** 2021-09-21

**Authors:** Johanna Ohlendorf, Luisa Stasch, Ulrich Baumann, Christiane Konietzny, Eva-Doreen Pfister, Gundula Ernst, Karin Lange, Kirsten Sautmann, Imeke Goldschmidt

**Affiliations:** 1Pediatric Gastroenterology, Hepatology and Liver Transplantation, Hannover Medical School, 30625 Hannover, Germany; stasch.luisa@mh-hannover.de (L.S.); Baumann.U@mh-hannover.de (U.B.); konietzny.christiane@mh-hannover.de (C.K.); pfister.eva-doreen@mh-hannover.de (E.-D.P.); goldschmidt.imeke@mh-hannover.de (I.G.); 2Medical Psychology, Hannover Medical School, 30625 Hannover, Germany; Ernst.gundula@mh-hannover.de (G.E.); lange.karin@mh-hannover.de (K.L.); sautmann.kirsten@mh-hannover.de (K.S.)

**Keywords:** pediatric liver transplantation, parental disease-specific training, education, discharge training

## Abstract

In the absence of widely accepted education standards for parents of children after liver transplantation (LTx), the content and structure of parental training are influenced by health care practitioners’ (HCP) individual knowledge and assessment of the relevance of its contents. This study examines the hypothesis that expectations towards training differ between HCPs and parents, and that the quality of parental training affects the job-satisfaction of HCPs. Attitudes towards disease-specific education were assessed by tailor-made questionnaires in HCPs (*n* = 20) and parents of children with chronic liver disease or after LTx (*n* = 113). These were supplemented by focused interviews in *n* = 7 HCPs and *n* = 16 parents. Parents were more satisfied with current counseling than HCP. Language barriers and low parental educational background were perceived as obstacles by 43% of HCPs. The quality of parental knowledge was felt to have a strong influence on HCPs job satisfaction. The expectations towards the content of disease-specific education largely overlap but are not synonymous. HCP and parents agreed with regards to the importance of medication knowledge. Parents rated the importance about the meaning of laboratory values and diagnostic procedures significantly higher (3.50 vs. 2.85, *p* < 0.001 and 3.42 vs. 2.80, *p* < 0.001) than HCPs. Parents and HCPs would prefer a structured framework with sufficient staff resources for disease-specific counseling.

## 1. Introduction

Structured educational interventions are key to the successful management of many different chronic diseases in childhood [1,2,3,4,5]. For diseases like type 1 diabetes, atopic eczema or asthma, the beneficial impact of structured disease-specific intervention programs on somatic outcomes and health-related quality of life has been repeatedly demonstrated [1,2,4]. Similar data are scarce in the context of pediatric liver transplantation.

Lerret et al. have examined whether pre-discharge education and the parental perception of readiness for discharge are associated with a reduction in coping difficulties and health-care utilization in children after solid organ transplantation [6]. They showed that parental perception of the quality of care coordination was associated with parental readiness for hospital discharge, which in turn was related to the family’s capacity to cope at home [6]. A qualitative analysis of parental expectations identified medication knowledge and the identification of complications as the most important topics for discharge teaching [7].

Pre-existing health-literacy, defined as “the degree to which individuals have the capacity to obtain, process, and understand basic health information and services needed to make appropriate health decisions” [8], is likely to impact onto the success of any existing parental training approach. Parental health literacy has been shown to be associated with the frequency of biopsy-proven rejections in adolescents after pediatric liver transplantation [9]. Chisholm et al. have described the impact on the health literacy of the interaction of patient-level factors, such as education, culture, socioeconomic status and health beliefs, with healthcare provider factors such as models of communication and education resources [10]. Their model of health literacy in transplantation (HeaL-T) emphasizes the need to address the mutual influence of patient and caregiver factors when designing an educational intervention for parents of transplanted children [10].

One lesson learned from pediatric diabetes care is that the unanimity of the health care team regarding therapeutic goals is more relevant for the metabolic outcome than the choice of insulin regime [11]. Good metabolic control in childhood diabetes is strongly associated with the health care professional’s agreement over metabolic target levels [12]. In the context of liver transplantation, targets for follow-up care are more difficult to define. While medication adherence for immunosuppressive therapy is an obvious choice for parental training, lifestyle topics such as risk behaviour, nutrition, UV protection, and physical activity are equally relevant for the reduction of long-term morbidity and mortality. In addition, long-term targets such as psychosocial rehabilitation and quality of life are gaining more and more attention [13,14,15]. In the absence of widely accepted education and training standards for parents of children after liver transplantation (LTx), the content and structure of disease-specific training are thus influenced by the health care professional’s individual knowledge, expectations, and preferences. This carries the risk of inadvertently falling short of parental needs. The purpose of this study was to examine health care professionals’ and caregivers’ attitudes towards the content of disease-specific education. We hypothesized that expectations and satisfaction with disease specific education will differ between parents and health care professionals (HCP), and that the quality of disease-specific training will affect the job-satisfaction of HCPs. Parental and HCP expectations, and the satisfaction with current disease-specific education, were examined using a combination of tailor-made questionnaires and focused interviews in a cross-sectional approach. The overarching aim of this study was to identify the obstacles for parental disease-specific education that need to be overcome for a successful long term outcome of pediatric liver transplantation.

## 2. Materials and Methods

Attitudes towards disease-specific education (the content, mode of delivery, and satisfaction with current practice) were examined using tailor-made questionnaires. In addition, focused interviews were conducted in a subset of parents/caregivers and HCPs in order to elicit a more thorough qualitative assessment of relevant attitudes.

### 2.1. Parents/Caregivers

All parents of patients aged 0–18 years followed in our liver unit between September 2019 and February 2020 for advanced chronic liver disease, or after liver transplantation were deemed eligible for study inclusion. Figure 1 summarizes the enrolment process. Only one parent per patient was included. Exclusion criteria were the lack of an ability to understand the German language on the part of the parents/caregivers, missing informed consent, or failure to complete the questionnaires. 113 parents/caregivers of 113 patients (68 girls, 45 boys) aged 6 months to 18 years (median 8.7 years) with chronic liver disease (*n* = 25) or after liver transplantation (pLTx) (*n* = 88) completed questionnaires. Participants’ demographic data are given in Table 1. A subset of 16 parents/caregivers participated in focused interviews.

### 2.2. Health-Care Professionals

All HCPs (doctors, transplant nurses, ward nurses, and dieticians) of the pediatric liver unit were invited to participate (Table 2). All doctors and at least 1/3rd of the nursing team (total *n* = 20) completed the tailor-made questionnaires. A subset of seven HCPs also participated in the focused interviews.

### 2.3. Questionnaires and Focused Interviews

Questionnaires addressed the general satisfaction with current disease-specific education, sources of health-related information, the importance of different knowledge topic areas, and the assessment of current parental knowledge. An English transcription of the tailor-made questionnaire is found in Section A.1 and Section A.2 in Appendix A. Questionnaires included a 17-item list of topics of disease-specific knowledge. For each item, parents and HCPs were asked to assess the degree of knowledge parents possessed in this particular topic, and to rate the importance of this particular topic. In order to evaluate their knowledge in relation to the corresponding importance of each item, the weighted knowledge was calculated analogously to weighted life satisfaction as described by Westergren et al. [16] and Ferrans et al. [17].

Focused interviews consisted of a series of open-ended questions on satisfaction, information preferences, and both the parental and HCPs perspective on the current practices of information-giving. Interview questions are listed in the Section A.3. During the interview, notes were taken which were then transcribed into a full report for each interview. Interview reports were reviewed independently (L.S., J.O., I.G.), and answers were grouped into categories of responses. Overlapping responses were reviewed together and the common answers and themes were identified.

### 2.4. Statistical Analysis

Baseline characteristics are described as numbers with frequencies for categorical variables and as median with interquartile range or mean with standard deviation for continuous variables. Missing data were not imputed. Likert scale data from questionnaires are presented as mean values of attributed numerical scores and were compared between parents and HCPs using *t*-test. For interview data a semi-quantitative descriptive analysis was performed. Statistical analysis was performed with SPSS Statistics 26 (IBM).

### 2.5. Ethical Considerations

This study was approved by the local Ethics Committee (Statement No. 8474) and was in accordance with the Helsinki Declaration on medical research involving human subjects. Informed consent was obtained from parents, caregivers, and HCPs.

## 3. Results

### 3.1. Satisfaction with Current Disease-Specific Education and Counseling

Both parents and HCPs gave good overall ratings to current counselling practice in our clinic. However, parents rated current counseling practice significantly better than HCPs (3.9 vs. 3.5 (*p* = 0.04) on a Likert scale from 1: very bad to 5: very good) (Figure 2). Parents were also more satisfied with current counseling practice (4.96 vs. 4.25, *p* < 0.01, on a 6-point Likert-scale). Only 3.6% of parents stated they were “very dissatisfied” or “rather dissatisfied” with current counseling, whereas 10% of HCPs rated themselves as “rather dissatisfied” (Figure 3). Results from the focused interviews reflected these findings: all parents declared themselves satisfied or very satisfied, while HCPs found the status quo predominantly “mediocre” or “with room for improvement”.

When asked about negative experiences or reasons for dissatisfaction with counseling, the following common themes emerged from parents’ interviews:(a)*Infrastructural problems*: lack of time, having to wait for scheduled counseling sessions, lack of staff and busy schedules on the ward as hindrances for successful counseling.(b)*Content of training*: poor background knowledge and difficulties in understanding medical terminology.(c)*On your own:* at least 25% of parents complained that they “often had to seek information on one’s own initiative”.

In HCP’s interviews, a “lack of dedicated time for sufficient parent education” (71%) and a “lack of standard protocols for parental education” (57%) were the most frequently mentioned drawbacks of the current counselling practice. HCPs also felt that language barriers and variations in parental educational background strongly affected counseling (43%).

40% of HCPs felt that parental disease specific knowledge affected their own work strongly or very strongly, while only 10 % felt no effect at all. In interviews, the proportion of HCPs who felt a strong impact of parental knowledge on their daily work increased to 75%. Almost half of the interviewees spontaneously mentioned that the degree of parental knowledge also affected their job satisfaction. The same proportion of HCPs felt that time constraints and a lack of staff will prevent any improvement in parent education.

### 3.2. Sources of Information

Hospital doctors (94%), nurses (46%), social workers (37%), and doctors in outpatient care (36%) were the most frequent sources of disease-specific information for parents. Family and friends (20%), psychologists (9%), therapists (8%), and support groups (6%) appear to contribute less to parental information. These numbers largely overlap with HCPs’ perception of sources of information. Parents would like to get more information from their health insurance (34%) and authorities or agencies (24%). In interviews, contact with other parents was mentioned as an important and sorely missed source of information by 44% of parents.

### 3.3. Success of Current Counseling Practice

Overall, parents rated their current disease-specific knowledge as “good” (Table 3). Knowledge on medication and “everyday life with the illness” scored best (4.09 and 4.04 on a 5-point Likert scale). Knowledge on psychological support options and social support options was rated as mediocre (2.94 and 2.09 respectively). The assessment of parental knowledge by parents and HCPs largely overlapped, albeit with some notable exceptions: HCPs perceived a lower level of parental knowledge than parents themselves in the areas of basic liver functions and anatomy, medication, diagnostic procedures, and everyday life with the illness (Table 3). In interviews, HCPs felt that parents were well informed about daily proceedings on the ward. Medication knowledge was perceived as patchy; 43% felt that parents were not sufficiently prepared for daily life after pLTx. No unifying theme regarding knowledge quality emerged from parental interviews, and 25% reported insufficient basic medical background knowledge.

### 3.4. Importance of Different Knowledge Topics

Importance of different topics of disease-specific knowledge was assessed using a 4 point Likert scale. Details can be found in Table 4. Knowledge on (immunosuppressive) medication and knowledge on the children’s primary disease were identified as the most important topics by both parents and HCPs. Some notable differences between parents and HCPs were documented: Parents had significantly higher ratings for importance in knowledge areas that enhance their understanding of illness and medical procedures. Specifically, they attributed more importance than HCPs to the anatomy of the liver, the meaning of laboratory values, and understanding diagnostic procedures. In contrast, knowledge on psychological support options was deemed more important by HCPs than by parents.

In order to compare knowledge in a specific topic in relation to the importance this particular topic held for parents and HCPs, we calculated weighted knowledge. Details are given in Table 5. Weighted knowledge by parents was highest in medication knowledge, immunosuppressive medication, and everyday life with the illness.

Results of the questionnaires are partly reflected in interviews: HCPs identified understanding the child’s illness, transplantation, life at home, and medication knowledge as the most important content of their counselling. Knowledge on diagnostic interventions or on basic liver functions was mentioned only infrequently. Parents differentiated between information they might have found useful before or immediately after transplantation, and knowledge they need further down the line. In contrast to questionnaire results, medication knowledge was raised as an important content by 2 of 16 parents only. The common themes identified for counseling after transplantation were (*a*) *daily life with a transplanted child*—recognizing and avoiding risks, recognizing complications, how to deal with schooling and leisure activities; (*b*) *planning for the future*: puberty, transition, higher education, job qualifications; and (*c*) *how is my child?*—getting an accurate assessment of current medical problems and state of health of their child.

### 3.5. Perceptions of Improvement

Questions about the potential improvement of current counseling practice identified two unifying topics in parents: (*a*) *resources*, and (*b*) *communication with peers.* For (*a*) *resources*, most parental answers focused on improving the resource framework for counseling, rather than demanding altered content. Measures mentioned included more time, more staff, contact options by email and phone, and provision of printed or online information resources. For *communication with peers*, meeting other parents to exchange experiences was overwhelmingly felt to be an invaluable resource for increasing parental knowledge. Parents repeatedly suggested that any new structural framework for disease-specific counseling should provide opportunities for parents to meet with other affected families.

In staff interviews, the *structure of parental training* emerged as the dominant topic. The majority of suggestions centered on creating a predefined structure of dedicated time slots for parental disease-specific training, supported by adequate staff resources. Training during in-patient stay was to be followed by scheduled “refresher” counseling sessions in the mid- and long-term follow-up. In contrast, the structure of content was less frequently mentioned. About a third of staff interviewees suggested to improve counseling by introducing guidelines and discharge protocols.

## 4. Discussion

Structured disease education programs have become a valued and frequently used tool to improve somatic and psychosocial outcomes in a number of chronic childhood diseases [4,18,19,20]. Data on the acceptance and efficacy of such programs in the context of pediatric liver transplantation are still scarce. This study investigated parental and HCPs’ attitudes towards an unstructured approach of disease-specific education. The overarching aim of this study was to identify obstacles for successful parental disease-specific education.

An examination of parental and HCPs’ satisfaction with the unstructured approach towards disease-specific education yielded heterogeneous results: parents were predominantly satisfied with the training they received. They felt generally well informed. In contrast, HCPs were less satisfied with the current unstructured approach and identified major drawbacks. Parental responses are likely subject to bias: most parents in this study have only ever been treated at their current liver transplantation centre and are thus unfamiliar with other approaches towards parent education. In addition, one has to assume a high degree of social desirability affecting parental answers. The extent of this bias is impossible to determine. Individual statements in interviews suggest that parents were indeed prepared to take a critical view towards the care they received. Nevertheless, it appears prudent to treat the overwhelmingly positive parental echo with some caution.

25% of HCPs stated that quality of parental knowledge had a “sizeable” impact on their daily work, while 40% felt this influence to be “strong” or “very strong”. Given this degree of impact on their work satisfaction, it appears understandable that the HCP rating for the current training is more critical. Lack of time, language barriers, and limited parental intellectual capabilities were mentioned both as obstacles for successful counseling, and as general hindrance for their work. Language barriers, cognitive capacities, and education level have been repeatedly identified as obstacles for health-literacy in the literature [10]. The fact that this actively reported in the HCP interviews highlights the significance of reciprocal influence of patient level and healthcare provider factors as described in the HeaL-T model on health literacy [10]. A study on the implementation of a new training program for adult renal transplant patients reports similar results [21]. In this study, implementation of a new training program increased job satisfaction, particularly in nurses with less than five years of work experience.

Parental disease-specific education in our unit is currently informally incorporated into clinic visits and routine care during in-patient stays, and is supplemented by one dedicated pre-discharge meeting after pLTx. Nurses, doctors, and members of the social support team are involved in educational activities, with nurses providing the lion’s share of medication training and disease-specific skills. Both parents and HCPs highlighted drawbacks of the current counseling approach, and made suggestions for improvement. Parents as well as HCPs quoted the lack of time and staff as the biggest drawbacks. For parents, lack of time was associated with feelings of “being left to their own devices”, i.e., a sense of vulnerability. HCPs, in turn, linked lack of time for counseling with ineffectiveness of their training efforts. These findings emphasize the reciprocal influence of healthcare provider factors (in this case time resources and organization of training) and health literacy.

The lack of structure in the current approach was mainly deplored by HCPs, not by parents. This is reflected in the recommendations for improving parental training. Besides providing a clear guidance for content of teaching, the implementation of a structured framework might also support HCPs job satisfaction by reducing some of their workload. In the study by Urstad et al. [21], the new structured patient training program systematically moved some of the patient education towards the outpatient setting. The ward nurses were thus relived from some of their responsibilities, which in turn enhanced job satisfaction and provided mental stress relaxation [21].

Parents in our study put less emphasis on structure. This difference might result from the different perspectives and expectations of HCPs and parents. It might also stem from the way open questions were phrased. In published literature, parents of children after solid organ transplantation considered education as particularly helpful if it was “consistent, came from knowledgeable staff” and “was repetitive” [7]. A standardized instrument, such as the Quality of Discharge Teaching Scale (QDTS) [22], would potentially have yielded more differential information on parental preferences. However, to the best of our knowledge, no such instrument is currently available in a German translation.

Parents and HCPs largely agreed with regards to importance of different topics, albeit with some exceptions. Parents and HCPs agreed in rating knowledge on the illness of their child and medication knowledge as most important. This is consistent with published literature. In a detailed qualitative analysis of open-ended questions asked before and three weeks after hospital discharge in 37 parents of children having received solid organ transplantation, medication knowledge was identified as the most important topic to learn about before discharge [7]. In contrast, knowledge on basic liver anatomy, on the meaning of laboratory values and the understanding of diagnostic procedures, such as ultrasound, MRI scanning, or radiography, was assessed significantly more important by parents than by health care professionals. The importance of “basic knowledge” information also became apparent in interviews, where parents mentioned a “lack of background knowledge” as one of the drawbacks of current teaching approaches. This observation highlights another potential problem of an unstructured, non-standardized approach towards parent education: namely, the possible omission of topics which are of interest for parents, but are not considered relevant by the HCPs.

In order to examine whether the (self-) assessment of current parental knowledge corresponds to perceived importance or the respective topic, we calculated weighted knowledge. Weighted knowledge accentuated the pivotal role of knowledge on primary disease and medication, as degree of self-assessed knowledge and ascribed importance overlap. However, weighted knowledge also accentuates certain differences in ratings and assessment between HCPs and parents, namely in the aforementioned areas of basic liver functions and anatomy, and laboratory values. Lee et al. have examined discrepancies between perceived importance and actual teaching practice in the context of preoperative teaching in adult surgery [23]. In their study, nurses considered information on anesthesia and the actual operative procedure as most important preoperatively, but would not concentrate on these topics in their teaching efforts. One possible explanation offered by the authors was the fact that information giving on these particular topics might be seen as the prerogative—or the task—of other HCPs. A similar situation might explain our findings: more nurses than doctors participated in our study. Nurses might consider giving background information on lab values or basic liver function as the doctors’ responsibility, and concentrate on more practical matters such as medication knowledge. This finding emphasizes the need for a structured distribution of teaching tasks, to avoid falling short of parents’ needs.

Our study has several limitations. An established, standardized questionnaire to assess parental preferences and assessment of discharge teaching would have been preferable, but does not exist in German. In the absence of standardized instruments, the composition of a tailor-made questionnaire, in particular with regards to knowledge topics, can be biased by the same degree of subjectivity that might impact on parental counseling in the first place. Only one third of nursing staff were able to participate in the study, mainly due to time constraints. A larger HCPs sample might have provided a more valid database. For parents, the importance of different topic areas might change over time. It is conceivable that procedural and medication knowledge appears more important before and shortly after transplantation, while questions of everyday life are more pertinent during follow-up. While these variations over time were mentioned by some parents during the interviews, they were not captured by our questionnaire. Despite these limitations, we feel that our study provides some valuable insight into the different perceptions of parental disease-specific education.

Despite the lack of a structured approach towards parental disease-specific education, satisfaction with current counseling practice is high among parents of liver transplanted children. Medication knowledge is regarded as the most important topic of education by parents and health care professionals alike. HCPs underestimate parental desire to not only receive practical knowledge, but to understand their child’s disease and medical results. Time, staff resources, and a structured framework toward parental disease-specific education could improve both parent and HCPs satisfaction. The implementation of a new structured approach would need to address parental information preferences and overcome language barriers in order to be successful.

## Figures and Tables

**Figure 1 children-08-00827-f001:**
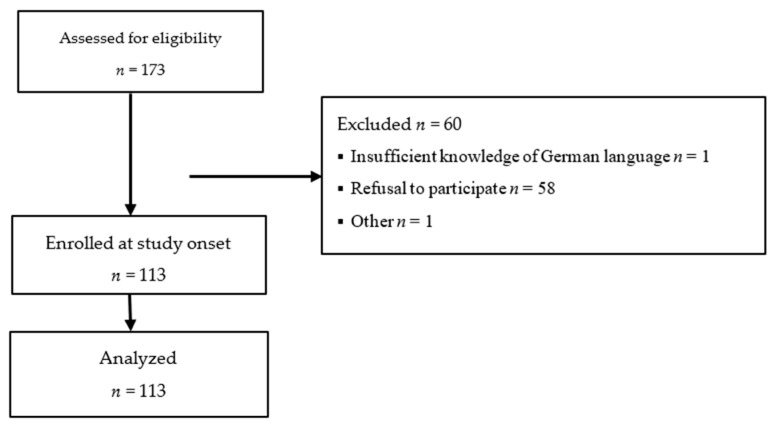
Patient enrolment and attrition.

**Figure 2 children-08-00827-f002:**
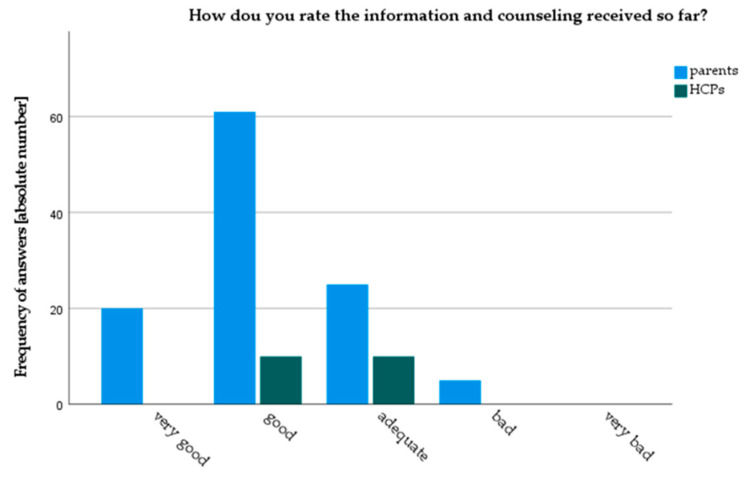
Rating of current counseling practice by parents and HCPs.

**Figure 3 children-08-00827-f003:**
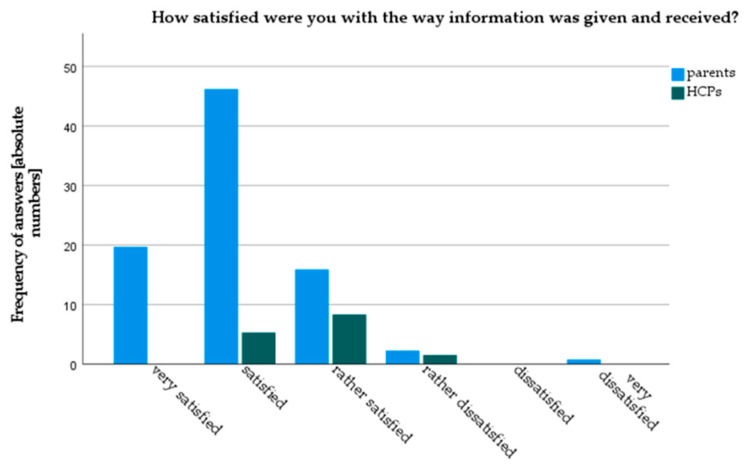
Satisfaction of parents and HCPs with current counseling practice.

**Table 1 children-08-00827-t001:** Characteristics of the study sample.

**Patients (*n* = 113)**	***n* (%)**
Sex	
Male	45 (39.8)
Female	68 (60.2)
Age at study entry (years)	
Mean SD	9.21 ± 4.87
Median, Min, Max	8.67, 0.50, 18.25
Transplantation status	
awaiting transplant	25 (22.1)
after liver transplantation	88 (77.8)
Age at transplant	
<1 year	38 (43.2)
1–4 years	32 (36.4)
5–12 years	17 (19.3)
13–17 years	1 (1.1)
Time since transplant (years)	
Mean (SD)	7.45 ± 4.53
Median Min, Max	7.17, 0.00, 17.00
Primary diagnosis	
Biliary atresia	70 (61.9)
PFIC	5 (4.4)
Acute liver failure	10 (8.8)
Metabolic	4 (3.5)
Alagille’s Syndrome	4 (3.5)
Tumor	6 (5.3)
Autoimmune Hepatitis	1 (0.9)
Other	13 (11.5)
**Parents (*n* = 113)**	***n* (%)**
Primary caregiver’s marital status	
information missing	4 (3.5)
Single-parent household	14 (12.4)
Two-parent household	95 (84.1)
Primary caregiver’s highest level of education	
information missing	4 (3.5)
No degree	2 (1.8)
General certificate	12 (10.6)
Secondary school	48 (42.5)
Highschool	25 (22.1)
University education	22 (19.5)
Native language	
German	82 (72.6)
other	27 (23.9)
information missing	4 (3.5)

**Table 2 children-08-00827-t002:** Characteristics of the HCP.

	Total Sample (*n* = 20)	Focused Interviews (*n* = 7)
Profession		
Physicians	8	2
Nurses	11	4
Dieticians	1	1

**Table 3 children-08-00827-t003:** Parental knowledge as assessed by parents and HCP.

Item	Parents		HCP		*p*
	mean	SD	mean	SD	
Level of knowledge					
Illness of their child	3.96	0.73	3.74	0.56	
Anatomy of the liver	3.30	0.83	2.65	0.67	0.001
Functions of the liver	3.42	0.76	2.85	0.67	0.002
Technical terms	3.17	0.93	3.15	0.67	
Proceedings during inpatient stay	3.77	0.99	3.42	0.69	
Transplantation	3.51	0.99	3.58	0.61	
Proceedings before LTx	3.37	1.10	3.30	0.66	
Proceedings during the inpatient stay for LTx	3.50	1.02	3.20	0.70	
Proceedings after LTx	3.55	1.10	3.20	0.70	
Meaning of laboratory values	3.42	0.92	3.00	0.56	
Medication of their child	4.09	0.74	3.30	0.80	<0.001
Immunosuppressants of their child	3.81	1.20	3.85	0.81	
Diagnostic procedures	3.75	0.79	3.05	0.89	<0.001
Everyday life with the disease	4.04	0.75	3.25	0.64	<0.001
Psychological support options	2.95	0.88	2.80	0.89	
Possible assistance for the care of their child	3.09	1.13	3.25	0.72	
Importance of nutrition	3.58	0.85	3.15	0.75	

Knowledge was assessed on a 5 point Likert scale ranging from 1: very little knowledge to 5: very good knowledge.

**Table 4 children-08-00827-t004:** Importance of knowledge as assessed by parents and HCPs.

Item	Parents		HCPs		*p*
	mean	SD	mean	SD	
Importance of knowledge					
Illness of their child	3.90	0.30	3.90	0.31	
Anatomy of the liver	3.31	0.59	2.89	0.32	0.003
Functions of the liver	3.36	0.54	3.15	0.37	
Technical terms	3.28	0.68	3.16	0.38	
Proceedings during inpatient stay	3.35	0.63	3.60	0.50	
Transplantation	3.69	0.50	3.75	0.44	
Proceedings before LTx	3.63	0.59	3.60	0.50	
Proceedings during the inpatient stay for LTx	3.59	0.55	3.60	0.50	
Proceedings after LTx	3.74	0.50	3.60	0.50	
Meaning of laboratory values	3.50	0.52	2.85	0.37	<0.001
Medication of their child	3.86	0.35	3.85	0.37	
Immunosuppressants of their child	3.79	0.56	3.90	0.31	
Diagnostic procedures	3.42	0.53	2.80	0.41	<0.001
Everyday life with the disease	3.74	0.44	3.60	0.50	
Psychological support options	3.14	0.75	3.60	0.50	0.008
Possible assistance for the care of their child	3.42	0.67	3.40	0.50	
Importance of nutrition	3.51	0.55	3.40	0.50	

Importance of knowledge was assessed on a 4 point Likert scale ranging from 1: not important to 4: very important.

**Table 5 children-08-00827-t005:** Knowledge weighted by importance as assessed by parents and HCPs.

Item	Parents		HCPs		*p*
	mean	SD	mean	SD	
Weighted knowledge					
Illness of their child	11.8	2.9	10.9	2.2	
Anatomy of the liver	9.1	2.7	7.0	2.0	<0.01
Functions of the liver	9.6	2.5	7.6	2.2	<0.01
Technical terms	8.7	3.1	8.6	2.1	
Proceedings during inpatient stay	10.6	3.6	9.6	2.6	
Transplantation	10.0	3.7	10.2	2.2	
Proceedings before LTx	9.6	3.8	9.2	2.3	
Proceedings during the inpatient stay for LTx	10.0	3.6	8.8	2.6	
Proceedings after LTx	10.2	4.0	8.9	2.6	
Meaning of laboratory values	9.4	3.7	8.1	1.6	<0.01
Medication of their child	12.2	3.0	9.2	3.2	<0.01
Immunosuppressants of their child	11.2	4.2	11.3	3.2	
Diagnostic procedures	10.6	2.8	8.2	2.6	<0.01
Everyday life with the disease	11.9	2.9	9.0	2.4	<0.01
Psychological support options	8.0	2.9	7.5	3.2	
Possible assistance for the care of their child	8.4	3.9	8.9	2.7	
Importance of nutrition	10.1	3.1	8.6	2.6	0.03

## Data Availability

Data are available from the corresponding author on request.

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
