# Peer review of "Two Sides of a Coin: Parental Disease-Specific Training as Seen by Health Care Practitioners and Parents in Pediatric Liver Transplantation"

_children, 2021, doi:10.3390/children8090827_

Round 1
Reviewer 1 Report
Thank you for the opportunity to review this manuscript, the topic is very relevant and I enjoyed reading the findings of the authors.
In general, I would feel the topic would benefit from being embedded in more detail in a model that would include for example details about health literacy, HCP-patient interaction, and self-management/adherence (e.g. PMID: 30464420 Chisholm-Bolms et al). Did the authors consider to incorporate pre-existing questionnaires about health literacy or patient reported outcomes or patient reported experiences in their work?
I have no other recommendations.
Author Response
Thank you for your thoughtful comments. We agree that the concept of health literacy and its many influencing factors is of paramount importance for disease-specific training in chronic childhood disease of any kind. We did not incorporate instruments assessing pre-existing or current health literacy into our study. The purpose of this study was to learn about parental expectations of health education, and about their experiences with the current system, rather than to investigate actual outcome of the current system. The study was driven by the hypothesis that ideas on adequate content of parental training might differ between HCP and parents, which in turn might lead to missing subjective parental needs.
Of course, a number of aspects concerning the reciprocal influence of patient level factors and healthcare provider factors on health literacy, as depicted in the HeaL-T model by Chisholm-Bolms et al mentioned in your comment, become visible in our study. We have therefore added the concept of health literacy in the introduction, and drawn a link to said model in the discussion where appropriate.
We are currently re-designing disease-specific education for both parents and patients of our paediatric liver transplant cohort. Implementation of the new training concept will be scientifically evaluated, and we are grateful for your comments as these have enhanced our conceptual framework for this task.
Reviewer 2 Report
Table, primary diagnosis; the sum (70+5+10+4+6+1+13=109) is not 113, 4 patients are missing?
Page 4, line 127: Did adolescents have to agree to have their parents participate in the study? After all, the adolescents did not participate in the study.
I would imagine, that the importance of knowledge (parents) depends on the age of the child at LTx and the period after transplantation. I would think that right after the transplant, understanding regarding the medications is more important than later on when "everyday life with the disease" comes to the fore. Such a change in the importance of knowledge over time (table 1: pre / post transplant, Time since transplant 0-17 years) or as a function of patient age (table 1: age at study entry 0,5 - 18,25 years) is not captured by this cross sectional study. This limitation should be added in the discussion.
Author Response
Thank you for your helpful comments.
4 patients with Alagille's Syndrome had gone missing from table 1 and have now been added again; this explains the missing numbers. Thank you for spotting this.
The sentence on consent was altered; consent was only obtained from the parents, not the patients.
We added a small paragraph in the discussion concerning the limitation of a cross-sectional approach with regards to changes in what is regarded as important over time. In fact, this phenomenon was actually mentioned by some parents during the interviews, but you are absolutely correct that the questionnaires did not capture this.